# Predicting Immunotherapy Efficacy with Machine Learning in Gastrointestinal Cancers: A Systematic Review and Meta-Analysis

**DOI:** 10.3390/ijms26135937

**Published:** 2025-06-20

**Authors:** Sara Szincsak, Péter Király, Gabor Szegvari, Mátyás Horváth, David Dora, Zoltan Lohinai

**Affiliations:** 1Translational Medicine Institute, Semmelweis University, 1094 Budapest, Hungary; vago.sara@phd.semmelweis.hu (S.S.); peter.kiraly@torokbalintkorhaz.hu (P.K.); szegvari.gabor@phd.semmelweis.hu (G.S.); matyiest@gmail.com (M.H.); 2Department of Anatomy, Histology, and Embryology, Semmelweis University, 1094 Budapest, Hungary; dora.david@semmelweis.hu

**Keywords:** machine learning (ML) models, linear regression, immunotherapy, response, survival, STAD, Gastric, colorectal (CRC) and other cancers

## Abstract

Machine learning (ML) algorithms hold the potential to outperform the selection of patients for immunotherapy (ICIs) compared to previous biomarker studies. We analyzed the predictive performance of ML models and compared them to traditional clinical biomarkers (TCBs) in the field of gastrointestinal (GI) cancers. The study has been registered in PROSPERO (number: CRD42023465917). A systematic search of PubMed was conducted to identify studies applying different ML algorithms to GI cancer patients treated with ICIs using tumor RNA gene expression profiles. The outcomes included were response to immunotherapy (ITR) or survival. Additionally, we compared the ML methodology details and predictive power inherent in the published gene sets using 5-fold cross-validation and logistic regression (LR), on an available well-defined ICI-treated metastatic gastric cancer (GC) cohort (*n* = 45). A set of standard clinical ICI biomarkers (MLH, MSH, and CD8 genes, plus PMS2 and PD-L1)) and de-novo calculated principal components (PCs) of the original datasets were also included as additional points of comparison. Nine articles were identified as eligible to meet the inclusion criteria. Three were pan-cancer studies, five assessed GC, and one studied colorectal cancer (CRC). Classification and regression models were used to predict ICI efficacy. Next, using LR, we validated the predictive power of applied ML algorithms on RNA signatures, using their reported receiver operating characteristics (ROC) analysis area under the curve (AUC) values on a well-defined ICI-treated gastric cancer (GC) dataset (*n* = 45). In two cases our method has outperformed the published results (reported/LR comparison: 0.74/0.831, 0.67/0.735). Besides the published studies, we have included two benchmarks: a set of TCBs and using principal components based on the whole dataset (PCA, 99% explained variance, 40 components). Interestingly, a study using a selected gene set (immuno-oncology panel) with AUC = 0.83 was the only one that outperformed the TCB (AUC = 0.8) and the PCA (AUC =0.81) results. Cross-validation of the predictive performance of these genes on the same GC dataset and an investigation of their prognostic role on a collated multi-cohort GC dataset of *n* = 375 resected, or chemotherapy-treated patients revealed that genes mannose-6-phosphate receptor (M6PR), Indoleamine 2,3-Dioxygenase 1 (IDO1), Neuropilin-1 (NRP1), and MAGEA3 performed similarly, or better than established biomarkers like PD-L1 and MSI. We found an immuno-oncology panel with an AUC = 0.83 that outperformed the clinical benchmark or the PC results. We recommend further investigation and experimental validation in the case of M6PR, IDO1, NRP1, and MAGEA3 expressions based on their strong predictive power in GC ITR. Well-designed studies with larger sample sizes and nonlinear ML models might help improve biomarker selections.

## 1. Introduction

One of the paramount challenges pertaining to the efficacy of immune checkpoint inhibitors (ICIs) lies in the identification of biomarkers across various malignancies that can consistently forecast the immunotherapy response (ITR) to ICIs, demonstrating favorable outcomes across multiple cohorts of cancer patients.

ICI treatment improved the outcomes of gastrointestinal (GI) cancer patients; however, a subset of patients with GC cannot benefit from these potentially toxic therapies [1,2,3]. The objective response rate of patients with GI cancers to ICIs was only 10–20%, and there is a need for effective biomarker-based patient selection [4]. Biomarkers with clinical applicability encompass microsatellite instability (MSI)/mismatch repair (MMR) status, a high tumor mutational burden (TMB), along with factors associated with the tumor microenvironment, including programmed death ligand 1 (PD-L1) expression and immune-related RNA expression [5,6,7]. Traditional biomarkers explain about 63% of the outcomes of ICI treatment, indicating that novel factors await discovery [8].

Machine Learning (ML) methodology can potentially accelerate ICI predictions using multi-omics data. To address the limitations of existing biomarkers, recent studies have begun to explore innovative approaches such as comprehensive genomic profiling and ML algorithms that can analyze complex datasets for more nuanced insights. For instance, a 24-gene RNA signature known as the IO-score has shown promise in predicting patient responses to ICIs, demonstrating significantly higher durable clinical benefit rates in patients with elevated scores compared to those with lower scores [7]. The designation ML encompasses a spectrum of methodologies, including classical interpretable linear models and non-linear models, which usually have better performance but limited transparency and explain ability, often referred to as black box models [9]. The most prevalent supervised ML methods are decision tree-based algorithms, support vector machines (SVMs), regression models (such as linear and logistic regression), Bayesian networks, neural networks, K-nearest neighbors (KNN), or ensemble methods, such as Random Forest (RF), or gradient boosting, such as XGBoost [10]. Nevertheless, identifying new biomarkers for robust and accurate prediction of immunotherapy response is highly warranted given the increasing body of multi-omics open access data available with variable or limited clinical data availability.

In the last decade, there have been significant advances in immuno-oncology and sequencing tests that identify targetable mutations and genes. Gene expression signatures are being actively developed to predict ICI outcomes. Aiming to complement these efforts with ML methods-based “-omics” layers, recent studies have begun to explore the utilization of transcriptomic data to guide the treatment of cancer patients [11,12,13,14,15,16,17].

This systematic review aims to assess the potential of RNA signatures as features in predicting immunotherapy response and to highlight their possible role in improving prognostic assessments. Furthermore, by evaluating and comparing methodological findings from various studies, our systematic review aims to critically evaluate how well ML algorithms can predict the effectiveness of immunotherapy in GI cancers by means of statistical model validation. Using synthesis, we compare the feature-selected gene sets with traditional clinical biomarkers using a unified modeling and validation approach.

## 2. Methods

### 2.1. Systematic Literature Review

The systematic search and study selection was conducted in accordance with the guidelines set out by the Preferred Reporting Items for Systematic Reviews and Meta-analyses (PRISMA) statement and the Cochrane Handbook for Systematic Reviews of Interventions [18,19].

### 2.2. Data Sources and Searches

Searches for relevant publications were carried out using the PubMed database. The search keywords included: (“immunotherapy” OR “ICI” OR “immune checkpoint”) AND (“machine learning” OR “deep learning” OR “artificial intelligence”) AND (“cancer” OR “tumor”) AND (“survival” OR “prognosis” OR “response” OR “predict”) + (“ai” [Title/Abstract] OR “ai” [Text Word] OR “ai” [Other Term]). The full definition of search keywords can be found in Appendix A.

### 2.3. Study Selection

The authors included publications that met the following criteria according to the PICO (Patient, Intervention, Comparison, Outcome) framework:Performed on gastric or colorectal cancer patients with RNA expression data and outcome reported; studies consisting of mixed gastric/colorectal and other cancer types were included if data on specific RNA gene expression and outcomes (survival, response to ICI therapy) and ML methods was reported specifically for the gastric and/or colorectal population;Test or validation cohort receiving immune checkpoint inhibitor treatment (anti-PD-1 or anti-PD-L1, and/or anti-CTLA-4);Reporting clinical outcomes: data on response (RECIST) or progression-free/disease-free survival (PFS/DFS) or overall survival (OS) or recurrence-free survival (RFS).

The authors excluded clinical trials (where not already FDA-approved treatments are used), reviews, systematic reviews, meta-analyses, editorials, correspondences, case series, case reports, commentaries, letters to the editor, protocols, and conference abstracts. Animal, in vitro (cell line, cell culture), and ex vivo (organoid) studies, any only preclinical studies, studies with medical imaging (radiomics, PET-CT, CT, MRI) and digital pathology (histopathological, whole slide imaging), or studies with vaccination, were also excluded.

All abstracts and full articles were reviewed according to the eligibility criteria by two reviewers; any differences in opinion were resolved through consultation with a third reviewer and a consensus was reached.

### 2.4. Data Extraction and Risk of Bias Assessment

Two reviewers collected data from the full text of the included studies. The following information was extracted from each study: authors, publication year, tumor in training cohort(s), interventions in training cohort(s), tumor in validation cohort(s), immunotherapy drug, machine learning method(s), and outcomes (Table 1). Additional details were extracted for an overview of prediction models: sample size, train/test split, variable selection, ROC AUC, machine learning method, model type, statistical model validation methods, and software (Table 2). Two authors independently extracted data, and any disagreements or discrepancies were discussed and resolved through the assessment of study eligibility.

We used the Newcastle–Ottawa Scale (NOS) to evaluate the quality of the included studies regarding the selection and the comparability of the cohorts and the determination of outcome of interest for cohort studies [20]. Studies with a score of ≥6 out of 9 were included in further analyses. Evaluation of included studies reported by the NOS is shown in Appendix A.

### 2.5. Statistical Analysis

Studies included in the systematic review used a variety of datasets to train and test their methods and the reporting of the mathematical formulation of the methods was incomplete; thus, a direct comparison of their predictive power was not feasible. To compare the predictive power inherent in the selected gene sets, we have evaluated their performance, serving as the basis for the same methodology. In this meta-analysis, we constructed a predictive score using logistic regression based on a well-defined ICI-treated available dataset (*n* = 45) [21], restricted to the published gene signatures, the dependent variable being a response to immunotherapy (binary). We chose this method because most of the studies examined herein also reported that they constructed their predictive scores with linear model frameworks. Overall, two genes were not available in this dataset. In addition, as points of comparison, we have selected a gene set from traditional clinical biomarkers associated with response to immunotherapy (MLH1, MLH3, MSH2, MSH3, MSH4, MSH5, MSH5-SAPCD1, MSH6, PMS2, CD274, CD8A, CD8B, CD8B2) and also calculated predictions on principal components of the whole dataset (highest explained variance, so that sum is over 99%, 40 components). Gene sets are listed in Table 3.

To evaluate the predictive power inherent in the best-performing immuno-oncology gene set [7], a separate logistic regression model was constructed for every gene to investigate its predictive power concerning immunotherapy response. Each gene’s expression data served as the independent variable in distinct models, with the immunotherapy response as the dependent variable. Metrics included (1) regression coefficient values, which indicate the direction and magnitude of each gene’s influence on the immunotherapy response, and (2) Odds Ratios (ORs) which were calculated from the coefficient values to provide an intuitive measure of the effect size, representing the change in odds of a positive immunotherapy response with a one-unit increase in gene expression.

Elastic-net-penalized logistic regression was performed on clinical covariates (MSI, Mutational Load, Immune Signature, PD-L1 CPS) to address multicollinearity and identify their weighted influence on immunotherapy response. Coefficients from this model were noted. A logistic regression model was then fitted to the entire dataset, including both gene expressions and clinical covariates. This model assessed gene expressions’ predictive power for immunotherapy response, conceptualizing the influence of clinical covariates based on elastic net coefficients. The mutational load was adjusted to a binary variable for simplicity.

**Table 1 ijms-26-05937-t001:** Characteristics of included studies.

Study	Tumor Type (Train)	IT in Training Cohort	Treatment (Train)	Verification Cohorts	IT Drug	Outcomes	ML Methods
Liu et al., 2022 [22]	CRC	No	CHT, targeted	CRC, Melanoma, Urothelial	anti-PD-L1	OS, RFS, RR	LASSO, SVM, RF, XGBoost
Lu et al., 2020 [7]	CRC, Gastric, Esophageal + other ^1^ cancers	Yes	anti-PD-1/PD-L1, anti-CTLA-4	CRC, Gastric, Esophageal, andother ^1^ cancers	anti-PD-1/PD-L1, anti-CTLA-4	OS, PFS, DCB/NDB	SVM
Cheong et al., 2022 [1]	Gastric	No	CHT, surgery	Gastric	anti-PD-L1	OS, RR	NTriPath, SVM
Wei et al., 2022 [23]	Gastric	No	adjuvant CHT	Gastric, Melanoma, Urothelial	anti-PD-L1, anti-CTLA-4	OS	LASSO-Cox
He et al., 2022 [24]	Pan-cancer (TCGA) ^2^	No	CHT *	Pan-cancer (TCGA, ICGC, and others) ^3^, mice, normal	anti-PD1, anti-CTLA-4	OS	LASSO-Cox, RF
Tang et al., 2022 [25]	Gastric	No	CHT	Gastric, Urothelial	anti-PD1	OS, PFS, RR	LASSO-Cox
Zhou et al., 2021 [26]	Gastric	No	CHT	Gastric, Urothelial, Melanoma, Lung	anti-PD-L1	OS, RFS, RR	LASSO-Cox
Lee et al., 2021 [13]	Melanoma, Pan-cancer (TCGA) ^2^	Yes	CHT, anti-CTL, A-4	Gastric, CRC, Melanoma, Lung, and others ^4^	anti-PD1, anti-CTLA-4	OS, PFS, RR	Cox proportional hazard model
Zhao et al., 2023 [8]	Gastric, Melanoma	Yes	CHT, surgery, anti-PD1/anti-CTLA4	Gastric, Melanoma	anti-PD1, anti-CTLA4	PFS, RR	GCNN

* IT in melanoma anti-PD1/anti-CTLA4 26/442‘ IT in urothelial cancer anti-PD1/anti-CTLA4 2/411. ^1^ Tumor types were not further specified in the study. ^2^ Gastric, colorectal, esophageal, head and neck, bladder, breast, cervical, bile duct, lymphoma, kidney, liver, lung, ovarian, pancreatic, prostate, melanoma, thyroid, uterine cancers, neuroendocrine tumors, glioma, glioblastoma, lymphoma, mesothelioma, sarcoma, thymoma. ^3^ Gastric, colorectal, esophageal, bile duct, pancreatic, liver, lung, nasopharyngeal, thyroid, bladder, kidney, prostate, uterine, ovarian, cervical, melanoma, skin, brain, head and neck, oral, breast, bone cancers, lymphoma, chronic lymphocytic leukemia, chronic myeloid disorders, lymphoproliferative syndrome, rare pancreatic tumors, glioma, glioblastoma, pediatric brain tumors, neuroendocrine tumors, thymoma, sarcoma, mesothelioma. ^4^ Ovarian, breast, leukemia, multiple myeloma, liver, kidney, bladder cancers. IT: Immunotherapy, CHT: Chemotherapy, TCGA: The Cancer Genome Atlas, ML: Machine learning, CRC: Colorectal, OS: overall survival, PFS: progression-free survival, RFS: recurrence-free survival, DCB/NDB: durable clinical benefit/no durable benefit, RF: random forest, XGBoost: extreme gradient boosting, SVM: support vector machine, GCNN: convolutional graph neural network.

**Table 2 ijms-26-05937-t002:** Characteristics of used ML models, train/test cohorts, and validation methods.

Study	Sample Size	Train/Test Split	Variable Selection	ROC AUC	ML Method	Model Type	Statistical Model Validation Methods	Software
Liu et al., 2022 [22], stemness-related genes	432 (train)/184 (test)	split: 70/30	**PDs**: 26 stemness gene sets from StemChecker; **VS**: I. ROC-AUC > 0.65 genes selected (247); II. LASSO regression and feature importance (RF, SVM, XGBoost) results intersected (nine)	**No reported AUC on IT sets**, two pairs of TPR/FPR could not be estimated in IT cohort: AUC train set: 0.996; AUC test set: 0.965	logistic regression	binary classification model	internal: GOF; external: predictivity on test set	caret, R 4.0.5
Lu et al., 2020 [7], immune-oncology-related genes	72 (train)/24 (test)	split: 75/25; CV outer loop: 70/30; CV inner loop: 60/40	**PDs**: sequencing panel targets (395); **VS**: I. statistical filter; II. recursive feature elimination cross-validation (RFECV) (24)	AUC test set 0.74	SVM linear kernel (Python)	binary classification model	internal: GOF, robustness by (10 + 3)-fold nested CV; external: predictivity on test set	scikit-learn, Python 3.6
Cheong et al., 2022 [1], 32-gene signature, molecular subtypes	576	no train/test split	NTriPath top three gastric cancer pathway genes (32)	TPR = 0.74, FPR = 0.27 is reported for IT (can be calculated)	SVM linear kernel	multiclass classification models	internal: robustness, LOO-CV	LIBSVM 3.17, Matlab R2018a
Wei et al., 2022 [23], cancer stem cell-related genes	~300	no train/test split	**PDs**: I. OCLR- >mRNAsi; II. WGCNA to identify most correlated gene module with mRNAsi (2527); **VS**: LASSO-Cox regression	ROC AUC for OS 1, 3, 5 years (0.723, 0.697, 0.724) in GSE62254 train set. **Independent IT cohorts**: IMvigor210: OS 12 months: AUC 0.664, ITR: 0.736; GSE91061: ITR 0.667; PRJEB25780: ITR 0.816	Cox regression	regression model	internal: GOF, robustness by 10-fold CV in 1 k iterations of Coxregression; external: predictivity on independent set and independent IT cohort	WGCNA, glmnet, ConsensusClusterPlus, survminer, timeROC, rms, R 3.6.3
He et al., 2022 [24], glutamine metabolism immunotherapy response genes	6049 (train)/2606 (test)	split: 70/30	**PDs:** results of latest reports and MSigDB (118); **VS:** genes ranked by mean accuracy decrease and mean Gini index decrease in RF modeling and intersected (16)	ROC AUC for ITR on TCGA (0.9173, *n* = 6049) and test (0.9162, *n* = 2606) cohorts and in anti-PD-L1 patients (0.8696, *n* = 24)	LASSO logistic regression	binary classification model	internal: GOF; external: predictivity on test set and independent IT cohort if GMIRS low/high groups were defined	survival, Random Forest, R 4.2.1, 4.0.0
Tang et al., 2022 [25], Cu-binding protein-related genes	~400	no train/test split	**PDs**: human Cu proteome and TCGA–STAD intersected (51) then intersecting DEGs(cancer vs. normal) (31); **VS:** I. univariate Cox regression- >associated prognosis (16); II. LASSO Cox regression (10)	Figure 3. ROC AUC for OS: 5 years AUC = 0.75, Figure 4. independent sets: OS 1 year AUC = 0.67 Figure 9. nomogram, AUC OS 1, 3, 5 years: 0.68, 0.69, 0.8	Cox regression	regression model	internal: GOF; external: predictivity on independent cohorts	SPSS25, limma, R 4.1.0
Zhou et al., 2021 [26], senescence- related genes	~400	no train/test split	**PDs**: survival genes, DEGs (GC vs. normal), and senescence genes intersected; **VS**: LASSO Cox regression (6)	score + stage + age: OS 5 years AUC 0.794; ACRG independent cohort OS 1, 3, 5 years (0.805, 0.772, 0.745)	Cox regression	regression model	internal: GOF (nomogram); external: predictivity on independent cohort (nomogram), independent cohort response ratios	survminer, survival-ROC, rms, glmnet, R 3.6.1
Lee et al., 2021 [13], synthetic lethality/synthetic rescue	~6300	no train/test split	I. initial pool of drug target SR interactions; II. compact biomarker SR signatures for IT	ROC AUC calculations on independent test sets, Figure 6. AUC values not explicitly given, approx. 0.66 (melanoma)–0.86 (STAD); Figure 7. WINTHER, ITR, AUC = 0.72	stratified Cox proportional hazard model	regression model	external: predictivity on multiple independent IT sets and WINTHER trial	survival, R
Zhao et al., 2023 [8], attention scores	121 (Liu), 45 (Kim), 50 (Gide)	split: 80/20		AUC = 0.85	geometric deep learning (graph neural network (GNN))	classification model	internal: robustness: LOO-CV, 5-fold CV; external: predictivity on multiple independent IT test sets. sample size dependence study	PyTorch, lifelines, scikit-learn, Python 3.7.1

PDs: predefined gene sets, VS: variable selection, ROC: receiver operating characteristic, AUC: area under the curve, GOF: goodness of fit, LOO: leaveoneout, CV: cross-validation, IT: immunotherapy, ITR: immunotherapy response, TPR: true positive rate, FPR: false positive rate, RF: random forest, XGBoost: extreme gradient boosting, SVM: support vector machine, OCLR: one-class logistic regression, mRNAsi: mRNAstemness index, GMIRS: glutamine metabolism immunotherapy response score, DEG: differentially expressed genes, ACRG: Asian Cancer Research Group, SR: synthetic rescue.

**Table 3 ijms-26-05937-t003:** Gene sets published in the original studies and the context of their selection.

Source	Context	Genes
Liu et al., 2022 [22]	Stemness-related	*GFPT1*, *PTMAP9*, *MOGAT3*, *DPM3*, *S100A12*, *PGM5*, *FUT6*, *SEMA3C*, *ADAM33*
Lu et al., 2020 [7]	Immuno-oncology-related	*IDO1*, *CCL22*, *IL13*, *TNFSF9*, *IFITM1*, *IFITM2*, *STAT1*, *IL1B*, *TAP1*, *NRP1*, *STAT6*, *CD163*, *KREMEN1*, *VCAM1*, *CCL2*, *LAPTM5*, *M6PR*, *BAGE*, *MAGEA3*, *MLANA*, *BRCA2*, *CDKN2A*, *EFNA4*, *PTEN*
Cheong et al., 2022 [1]	Molecular subtype characterization	*ACTA2*, *AREG*, *ASCC2*, *BEST1*, *BRCA1*, *CREBBP*, *DDX5*, *EP300*, *ESR1*, *CIAO2A*, *FHL2*, *GNL3*, *HIPK2*, *HSF1*, *IGSF9*, *JUN*, *MSH6*, *NCOA6*, *TGS1*, *PARP1*, *PAWR*, *PCNA*, *PML*, *PPP2R5A*, *RPA1*, *SMAD3*, *SMARCA4*, *TP53*, *TP63*, *WRN*, *WT1*, *WTAP*
Wei et al., 2022 [23]	Stemness-related genes	*CLNS1A*, *DUSP3*, *FANCA*, *FANCC*, *H3C2*
He et al., 2022 [24]	Glutamine metabolism	*CTPS2*, *E2F3*, *EIF2A*, *EPAS1*, *JAK2*, *L2HGDH*, *MIOS*, *PPAT*, *PYCR1*, *SDHD*, *SEH1L*, *SIRT5*, *SLC38A5*, *TET1*, *TGFB1*
Tang et al., 2022 [25]	Cu-binding proteins	*AFP*, *ALB*, *CP*, *ENOX1*, *F5*, *LOX*, *LOXL3*, *SLC31A1*, *SNCA*, *SPARC*
Zhou et al., 2021 [26]	Senescence-related	*ADH1B*, *EZH2*, *IL1A*, *SERPINE1*, *SPARC*, *TNFAIP2*
Lee et al., 2021 [13]	Synthetic lethality	*CD27*, *IL15RA*, *TNFRSF13C*
Clinical markers	ICI efficacy-related traditional clinical biomarkers	*MLH1*, *MLH3*, *MSH2*, *MSH3*, *MSH4*, *MSH5*, *MSH5-SAPCD1*, *MSH6*, *PMS2*, *CD274*, *CD8A*, *CD8B*, *CD8B2*

The scikit-learn library (Python) was employed to implement logistic regression models. The results of the logistic regression, PCA and ROC analyses, along with the calculated metrics, were compiled in a Jupyternotebook (7.0.2.) which was placed ina Zenododata repository to demonstrate reproducibility. Due to the small sample size, a 1000-fold stratified 5-fold split was used. To reduce the error due to chance, all models reported in the article were evaluated on identical splits which were created manually from random permutations of the samples. For full details see the attached Jupyter notebook. The Kaplan–Meier (KM) Plotter online framework created by Győrffy was used to perform KM analysis in assessing the prognostic role of key genes identified in the IT-treated GC dataset of Kim et al. [21,27,28]. The KM plotter is freely accessible at https://kmplot.com/analysis/ (accessed on 4 January 2025).

## 3. Results

### 3.1. Literature Search

A comprehensive and systematic search of the PubMed database identified a total of 1125 abstracts. The search keywords included: (“immunotherapy” OR “ICI” OR “immune checkpoint”) AND (“machine learning” OR “deep learning” OR “artificial intelligence”) AND (“cancer” OR “tumor”) AND (“survival” OR “prognosis” OR “response” OR “predict”). The full definition of search keywords can be found in Appendix A. This search process involved a broad assessment of the available literature to ensure that a wide range of relevant research was included. Abstracts of the 1125 publications were screened based on the inclusion/exclusion criteria. The authors included publications that met the following specific criteria focusing on gastric or colorectal cancer patients with RNA expression data and reported outcomes. The authors allowed inclusion of studies with mixed cancer types if they provided specific RNA gene expression data and outcomes for gastric and/or colorectal populations. Key requirements included the presence of a test or validation cohort receiving ICI treatment (anti-PD-1, anti-PD-L1, or anti-CTLA-4) and reporting clinical outcomes such as response (RECIST), progression-free/disease-free survival (PFS/DFS), overall survival (OS), or recurrence-free survival (RFS). In order to properly compare studies, the initial broad search had to be narrowed down considerably during the screening process. After screening the abstracts, 701 abstracts were eliminated due to their lack of relevance. A total of 424 studies were retrieved for full-text screening, and 408 were also excluded due to irrelevance. Among the remaining 16 studies, five were excluded for ineligible study design and one for ineligible outcomes. Animal studies, in vitro and ex vivo experiments, as well as studies focused solely on medical imaging or digital pathology, were also excluded. Ultimately, nine studies fulfilled the selection criteria and were included in the systematic review. The detailed process of study selection is depicted in Figure 1.

### 3.2. Study Characteristics

The characteristics of the articles reviewed are summarized in Table 1, and their machine learning methodologies are described in Table 2. Among the nine included studies, five used training cohorts exclusively composed of patients with gastrointestinal (GI) cancers, including four that focused on gastric cancer (GC) [1,23,25,26] and one on colorectal cancer (CRC) [22]. The remaining four studies included two pan-cancer analyses [13,24], where the training cohorts encompassed patients with GC, CRC, and various other cancer types (detailed in Table 1). Furthermore, Lu et al. used a cohort primarily of GI cancer patients (*n* = 61) with a smaller subset (*n* = 11) from unspecified cancers, and Zhao et al. constructed their training cohort from gastric cancer (GC) and melanoma cases [7,8].

Regarding verification cohorts, most studies included GC (*n* = 8), melanoma (*n* = 6), and urothelial cancers (*n* = 6), with additional cohorts covering CRC (*n* = 4), esophageal (*n* = 3), and lung cancers (*n* = 3). Other cancers were also represented in smaller numbers (*n* = 2) (as detailed in Table 1). One study also included data from normal tissue in the verification cohort [24].

Immune checkpoint inhibitor (ICI)-treated cohorts primarily involved gastric cancer (*n* = 6), melanoma (*n* = 5), and urothelial cancer (*n* = 5), with additional inclusion of colorectal (*n* = 1), esophageal (*n* = 1), lung (*n* = 2), and kidney (*n* = 1) cancers.

Two studies included immunotherapy-treated patients directly in their training cohorts, specifically with anti-PD-1, anti-PD-L1, and anti-CTLA-4 therapies [7,8].

In the verification cohorts, anti-PD-1 or PD-L1 inhibitors were used in five studies each, and anti-CTLA-4 in five as well. While most TCGA datasets were from non-immunotherapy-treated cohorts, two datasets contained immunotherapy-exposed patients: TCGA-SKCM (melanoma; 24 patients) and TCGA-BLCA (urothelial cancer; two patients), both treated with anti-PD-1/CTLA-4.

All studies analyzed RNA-seq data from human tumor tissues, with He et al. (2022) [24] also incorporating mouse colon cancer data. Survival data was available in all studies: overall survival (OS, *n* = 8), progression-free survival (PFS, *n* = 4), and recurrence-free survival (RFS, *n* = 2) [24]. Response outcomes included response rate (RR, *n* = 6) and durable clinical benefit/no durable benefit (DCB/NDB, *n* = 1). Gene signatures were initially tested in either chemotherapy (*n* = 7) or immunotherapy (*n* = 2) training cohorts. Classification models were used in five studies, and regression models in four. All models were evaluated on immunotherapy-treated cohorts.

### 3.3. Biological Background of Predictive Gene Sets

Articles in the field mostly attempt to build upon already existing biological knowledge and codify it in terms of RNA expression data, rather than attempting to identify novel gene sets/pathway associations, therefore limiting the input data the ML algorithms can work with. We identified three approaches: (1) detection of TME (tumor microenvironment) cell types [1,22,23,26], (2) evaluating the activity of a given pathway [24,25], and (3) looking for directly treatment-related genes [7,8,13]. The works of Lee et al., Cheong et al., and Zhao et al. show a more generalized approach, establishing their gene set based on systematic screening rather than the association of a general trait with prognostic measures [1,8,13].

Of those that focused on describing cellular level qualities, the groups of Liu et al. and Wei et al. set out to predict the cancer stem cell content of the tumor [22,23]. The presence of such cells has been related to drug resistance and relapse after treatment [29]. Cheong et al. defined cancer subtypes based on pathway analysis and found that they differ with regard to OS [1]. Zhou and colleagues investigated the phenomenon of cellular senescence and found that it is more prominent in the TME, though not in the cancer cells themselves, and that its level is associated with OS and RFS [26].

Two studies focused on molecular-level qualities, namely glutamine metabolism and Cu-binding proteins [24,25]. Both studies’ primary aim was to create an index that describes the process of interest, and establishing a connection to prognosis was secondary. Glutamine metabolism is crucial for tumor cells and has been shown to affect antitumor T cell response [30]. Free Cu was discovered to mediate a process of cell death termed cuproptosis, and to be able to inhibit GC growth [31,32].

We found only three studies with a primary aim of predicting clinical outcomes in relation to immunotherapy. Lu et al. based their work on a predefined 395-gene immune-oncology panel [7]. The teams of both Lee and Zhao started their work from the molecular targets of current immunotherapeutic agents. Lee et al. examined the possibilities of combination therapies, by attempting to discover other genes that might rescue their loss [13]. Zhao et al. used the gene regulatory network neighborhood of PD1/PD-L1 to train an artificial neural network (ANN) to predict responder status [8]. Gene sets are listed in Table 3.

### 3.4. Sources of Transcriptomic Data

The transcriptomic datasets used for modeling have cohort sizes in the range of 45–6500 patients; only two studies used a cohort with a sample size below 100. This means that the possibility of overfitting due to a low sample size is avoided in most studies. In the studies of Liu et al., He et al., and Zhao et al., one dataset is split into a training and test set, and the model is learned from the training data, while the independent test data is used to numerically assess the predictive performance of the model [8,22,24]. Lu and colleagues use a 13-fold split of the dataset due to the nested cross-validation (CV) procedure for model validation [7]. Others utilize a full cohort for training the model without splitting and include a dataset from a separate source for model validation purposes (see Table 2).

We identified two groups of studies based on ML models. Studies in the first group limit the set of genes in the transcriptomic data by predefining the cancer-attributed process before variable selection. Liu et al. collected 26 stemness gene sets from the StemChecker webserver, He et al. combined results of the latest reports and gene sets of glutamine metabolism from MSigDB [33], Tang et al. investigated genes of the human Cu proteome [34], Zhou et al. collected three sets of genes—differentially expressed genes (GC versus adjacent), senescence gene sets from MSigDB, and survival genes—and Wei et al. used genes which correlate well with a stemness index [22,23,24,25,26]. In the study of Lu et al., a specific sequencing panel of 395 immune-related genes was used (see Table 2) [7].

In the other group, the genes are selected through an algorithmic search in a knowledge base. Cheong and colleagues identify gastric cancer-related pathways/genes with the NTriPath algorithm; Lee et al. search for genetic interactions of drugs with the SELECT algorithm [1,13]. The work of Zhao et al. also belongs to this group since no limitation is imposed on the number of genes (see Table 2) [8].

### 3.5. Variable Selection

Variable selection is applied just before the model is trained on the transcriptomic data, aiming to reduce the number of genes fed into the model optimization algorithms, both for practical reasons and to increase interpretability and to avoid overfitting, i.e., biasing the model toward the specifics of the training set. The selection process removes genes not associated with response or survival and eliminates highly redundant genes. The gene sets used by the reviewed studies and our benchmark are listed in Table 3.

In the studies that used computational variable selection, four different approaches can be identified, of which some are used sequentially. Zhou et al. utilize the intersection of three predefined gene sets, while Tang et al. intersect the Cu proteome with differentially expressed genes (gastric cancer vs. normal tissue) to reduce the number of genes. Lu et al. define the most extensive statistical filter by thresholding for variance, area under the curve (AUC), mutual information, and *p*-values for patient subgroups to reduce the number of genes [7,25,26]. Liu et al. filter genes by an AUC threshold, and in the study of Tang et al., only statistically significant genes are kept based on univariate Cox regression [22,25]. Liu et al. assign weights to the genes with LASSO regression, and determine feature importance with SVM, random forest, and XGBoost modeling [22]. The number of genes is then reduced by intersecting the most important genes from the four different algorithms. He et al. assign a mean decrease in the Gini coefficient and a mean decrease in accuracy to the genes by RF modeling [24]. In the studies of Wei et al., He et al., Tang et al., and Zhou et al., LASSO variable selection and regularization are used simultaneously for Cox regression [23,24,25,26]. Lu and colleagues use recursive feature elimination CV in the SVM modeling step [7]. Cheong et al. and Lee et al. do not apply statistical variable selection; their modeled gene sets are determined by search algorithms from knowledgebases (see Table 2) [1,13].

### 3.6. Modeling

The included studies follow two schemes. In the first one, risk groups are created in most cases based on the product of a Cox regression, usually termed the score. The score is a linear combination of gene expression values. Following the calculation of the score for all patients, the cohort is split based on the median or the sign of the score into two or more groups. Kaplan–Meier analysis is then used to visualize the difference in survival between the stratified groups. This scheme is used by four studies [23,24,25,26]. In the study of Lee et al., the predictive score is defined as the ratio of downregulated genes identified by their SELECT algorithm, which also incorporates Cox regression [13]. In the studies of Liu et al. and Cheong et al., the risk groups are established by a clustering algorithm and, sequentially, a classification model (logistic regression and SVM with linear kernel, respectively) is trained to predict the cluster membership [1,22]. The predicted clusters are assessed by Kaplan–Meier analysis or the response rate. This scheme is applied in 7/9 studies and establishes an indirect connection to survival or ITR by linear modeling.

In the second scheme, the response to immunotherapy is directly modeled (2/9 studies). Zhao’s group trains a nonlinear graph neural network (GNN) classifier model. Lu et al. use an SVM classifier with a linear kernel to discriminate between responders and non-responders; a distance-based score is defined. Based on the Youden index, the cohort is split into risk groups; sequentially, responder rate is assessed, and survival is based on Kaplan–Meier curves [7,8].

### 3.7. Validation

The validation techniques used are summarized in Table 2. OECD guidelines for computational modeling define internal and external statistical model validation. In internal validation, the aspects of goodness of fit and robustness of the model are assessed numerically by validation parameters, while in external validation model predictivity is estimated [35]. In the case of linear methods, all three aspects should be considered, while for ensemble methods, support vector machines, and neural networks, the goodness of fit is of reduced interest due to their high flexibility in fitting the train set. Table 2 shows which aspect is assessed by the included studies.

All studies assessed internal validation. In the case of Lee et al., the internal validation of their SELECT algorithm is presented in their previous study [36]. The studies of Liu et al., He et al., Tang et al., and Zhou et al. do not assess the robustness aspect of internal validation although such metrics are indicative of the stability of the model given the sample size and the possibility of overfitting [22,24,25,26]. External validation is straightforward if the dataset is split prior to modeling and the final model is validated on the test part, which was not used in modeling, as shown by the studies of Liu et al., Lu et al., He et al., and Zhao et. al. [7,8,22,24]. Some studies define risk groups and present the comparison of those in independent cohorts as validation. We refer to the study of Royston et al., who discuss the pitfalls of such discrimination [37].

### 3.8. Generalization of Gene Set Importance

Using logistic regression, we compared the gene sets published in the studies to create a prognostic score. We used a uniform methodology and evaluated them on the same dataset using ROC analysis (Figure 2).

Three studies show a clear drop in predictive power, including the two with the highest reported ROC AUC values on their test/validation datasets (reported/comparison: He et al.: 0.9162/0.689; Zhou et al.: 0.805/0.624; Lee et al.: 0.72/0.606) [13,24,26]. Our method nearly replicated the results of Wei and colleagues [23]. In two cases our method has outperformed the published results (reported/comparison: Lu et al.: 0.74/0.831; Tang et al. 2022: 0.67/0.735) [7,25]. Besides the published studies, we have included two benchmarks: a set of traditional clinical biomarkers (MLH, MSH, CD8 genes, PMS2, and PD-L1) and using principal component analysis based on the whole dataset (PCA, 99% explained variance, 40 components). Interestingly, Lu et al.’s selected gene set (immuno-oncology panel) (AUC = 0.83) was the only one that outperformed the clinical benchmark and the PC results [7]. Therefore, we decided to evaluate one-by-one the predictive power of these genes on the immunotherapy-treated GC cohort of Kim et al.to identify potential individual biomarkers [21].

### 3.9. Predictive Value of Key Genes

We performed logistic regression with 5-fold cross-validation on the Kim et al. sequencing dataset and extracted the Odds Ratio (OR), coefficient value, and classification accuracy metrics for all features from the immuno-oncology gene set of Lu et al. to reveal the predictive power of every gene, not just the ensemble gene set, on ITR [7,21]. Of note, Kim et al. in their study evaluated only complex signatures (EMT (epithelial-mesenchymal subtype) signature, proliferation score, immune signature) and established clinical markers (MSI, Mutational load, PD-L1 CPS), but not individual genes [21]. Univariate analysis showed that genes mannose-6-phosphate receptor (M6PR), Transporter 1, ATP-Binding Cassette (TAP1), TNF Superfamily Member 9 (TNSF9), Signal transducer and activator of transcription 1 (STAT1), Ephrin A4 (EFNA4), Interleukin-13 (IL-13), Interferon-induced transmembrane protein 1 (IFITM1), and Indoleamine 2,3-Dioxygenase 1 (IDO1) showed a significant effect with OR > 2 towards response, whereas STAT6, Neuropilin 1 (NRP1), Protein melan-A (MLANA), MAGE Family Member A3 (MAGEA3), and Vascular cell adhesion protein 1 (VCAM1) showed a significant effect with OR < 0.8 towards non-response (Figure 3A) Of note, EFNA4 and IL-13 performed with only 71% and M6PR with only 64% accuracy in correctly classifying cases into responders and non-responders (Appendix A) that is also reflected in their large confidence interval. Moreover, none of the genes showing significant effect towards non-response displayed greater accuracy than 71% (Appendix A). In contrast, TNFSF9 and IDO1 performed with an accuracy of 93% and 86%, respectively (Appendix A), exhibiting relatively narrow confidence intervals.

To avoid the interpretation of spurious effects from potential multicollinearity, we implemented the most important clinical covariates from the metadata of Kim et al., including MSI status (high vs. low), mutational load (high, moderate, low), PD-L1 combined positive score (CPS), and immune signature (high vs. low) [21]. First we fitted an elastic-net-penalized logistic regression model, with a custom penalty mix, penalizing only the clinical covariates. As this data is not high-dimensional, we chose this standard method for simplicity. We are aware that for full transcriptomic datasets, the effectiveness of LASSO and elastic net variable selection is being questioned [38]. As expected, both PD-L1 CPS and MSI were among the five best predictors, whereas immune signature did not show a high coefficient level (Figure 3B). The model showed that M6PR, IDO1, CDKN2A, TNFSF9, and IL-13 had the strongest effect on survival from the signature genes, M6PR outperforming all established clinical markers, IDO1 outperforming all clinical markers except for PD-L1 CPS, and CDKN2A outperforming mutational load when predicting ITR. In predicting non-response, none of the clinical markers showed a strong effect; in contrast, genes VCAM1, MAGEA3, and NRP1 remained among the strongest predictors of non-response in the multivariate setting as well (Figure 3B).

Due to the lack of another independent immunotherapy-treated GC cohort, externally validating the predictive role of the aforementioned genes was not feasible. However, the prognostic roles of the same genes were tested on RNA-seq datasets of publicly available surgically resected and/or chemotherapy-treated GC cohorts. For this purpose, a comprehensive Kaplan–Meier plotter tool was published earlier by Győrffy, concatenating RNA-seq- and clinical data from GEO, EGA, TCGA, METABRIC, Impact, and PubMed repositories from a total of *n* = 375 GC patients [27]. According to the plotter, M6PR-high patients show a strong trend towards increased survival [HR:0.64 (0.45–1.01), *p* = 0.052], when including patients from all stages; and a significantly increased survival in the case of advanced-stage (III–IV) GC patients [HR:0.63 (0.42–0.95), *p* = 0.026]. High expression of IDO1 is also associated with a trend towards increased survival [HR:0.72 (0.51–1.01), *p* = 0.054], whereas TNFSF9 [HR:1.39 (0.95–2.05), *p* = 0.091] and CDKN2A [HR:0.77 (0.56–1.06), *p* = 0.11] show no significant association with survival. IL13-high expressors show decreased survival time [HR:1.49 (1.06–2.1), *p* = 0.02] (Appendix A). High NRP1 [HR:2.13 (1.51–3.01), *p* < 0.001] and MAGEA3 [HR:1.65 (1.35–2.01), *p* < 0.001] expressions are both associated with significantly decreased survival time, in contrast with VCAM1 that shows no significant association with survival time [HR:0.88 (0.75–1.05), *p* = 0.15] (Appendix A).

## 4. Discussion

The administration of ICIs has improved outcomes in GI cancers; however, patients’ objective response rate varies greatly, and patient selection based on traditional clinical biomarkers provides benefits to a certain extent [4]. ML methods hold the potential to accelerate the development of biomarkers in immuno-oncology. In this systematic review, we identified various methodologically distinct attempts at establishing prognostic scores based on the mRNA expression of selected gene sets.

An increasing number of computational models for GI malignancies attempt ICI outcome prediction. However, progress is hindered by the low number of open-access transcriptomic datasets on gastrointestinal cohorts that received immunotherapy (Table 1 of Kovács & Győrffy) [39]. Additionally, the small sample size (<50) of the cohorts is also a limiting factor, due to their tendency for overfitting. Thus, most studies opt for using larger datasets without IT as training sets to circumvent the limitations of sample size and data availability. Open-access TCGA datasets like colon and rectal adenocarcinoma (COAD + READ) [22], stomach adenocarcinoma (STAD) [25,26], TCGA pan-cancer expression profiles [24], or Gene Expression Omnibus datasets are popular choices [1,23]. IT response is then only inferred by evaluating the obtained model on immunotherapy sets, weakening predictive power by using an indirect methodology. In our study, we found nine studies using ML methods to characterize GI cancers. While all of them published conclusions regarding immunotherapy, only two utilized IT cohorts for model training purposes: those from the groups of Zhao and Lu [7,8].

Most studies investigate cancer-related phenomena or processes (e.g., cancer stemness [22,23], cellular senescence [26], or glutamine metabolism and use them as proxies to make statements about ITR [24]. In these studies, the main interest is to generate risk stratifications in the cohorts based on RNA expression data and identify descriptive gene subsets. The stratified groups are then visualized on Kaplan–Meier curves to showcase the power of the model. We have found only three studies that aimed to directly predict ITR: Lu et al., predicting durable clinical benefit (DCB), based on RECIST, Lee et al., who compared the response rate predicted by their score to independent studies, and Zhao et al., who predicted responder status [7,8,13].

The most common techniques used to obtain prognostic indices were linear methods, namely the Cox proportional hazard model and logistic regression. Both models use a prognostic index obtained as a linear combination of the expression of a gene set. In the case of the Cox model, this is inserted as the exponent to estimate the hazard function [13,23,25,26]. In logistic regression, the prognostic index is transformed into a probability value [22,24]. Support vector machines (SVMs) are of great interest due to their power to find complex, nonlinear solutions for separating groups of interest. However, only two studies use SVMs and only with a linear kernel, meaning that the definition of the decision boundary is still linear [1,7]. In the articles we reviewed, nonlinear, ensemble methods (specifically XGBoost and the RF algorithm) are used only to generate feature importance values for variable selection, i.e., the definition of gene sets prior to modeling [22,24]. Of exceptional interest is the study by Zhao et al., who developed a graph neural network incorporating gene regulatory network data that handles the direct prediction of immunotherapy response as a nonlinear problem [8].

Most studies that our systematic search found have calculated predictive scores based on gene sets selected from previously published gene expression signature data that were not assembled to be directly descriptive of response or survival. This restriction on the genes available to the ML methods would introduce a strong bias to any attempt at pathway analysis across studies. In addition, exact mathematical formulation of the prognostic scores was not available in most cases, hindering proper comparison. Thus, we strongly advocate for stricter standards in the publication of computational methods, requiring reproducibility by independent researchers.

Some of the included studies have notable limitations. The group of Wei has chosen their training set from multiple candidates based on a calculated score showing association with outcomes [23]. This creates a form of selection bias at an early step of their analysis. It is clear that Tang and colleagues have applied mathematical transformations to at least two sets of the ROC curves they published, without specifying these modifications [25]. In their article, Figure 3 contains no monotone curves, and those in Figure 9 have been visibly smoothed. While Liu et al. do use nonlinear methods for variable selection, there is no mention of hyperparameter optimization or checking the robustness of such results [22]. In addition, they then settle on an intersection of gene sets, including ones that were defined by linear techniques, which could easily lead to the exclusion of those genes that only the nonlinear methods could identify. In the works of Liu et al., and Cheong et al., the target variable of the modeled dataset is assigned by cluster analysis [1,22]. It is demonstrated on Kaplan–Meier curves that the predicted patient clusters in the independent cohorts have significantly different survival. In the work of Cheong et al., a multiclass classifier model is used to recreate the clustering in another patient cohort. Although in both studies there is data supporting the correctness of the models, there is a possibility of residual confounding, and it should be debated whether external validation of a model where the target variable is not present in the independent test set is possible [37].

While we were not able to reproduce the results of the articles examined due to the inaccessibility of raw data and thus proper methodological comparison was limited, we still chose to attempt a measure of comparison by examining the predictive power of the gene sets alone. Since most used linear methods, we applied logistic regression, expecting comparable results. This we obtained in only a single case [23]. In the case of the works of He, Zhou, and Lee, the selected gene sets underperformed both the reported ROC AUC and our benchmark sets by a considerable margin [13,24,26]. The groups of He and Lee reported only the use of a form of Cox regression based on the gene set [13,24]. While there are differences between linear methods, based on the magnitude of discrepancy we observed, we recommend investigating these models’ generalizability. In two cases, our method outperformed the published ROC AUC values. One of those was the work by Tang and colleagues [25]. As they did not set out to directly predict immunotherapy response, but levels of Cu-binding proteins, the more focused construction of our model might explain this phenomenon. Regarding the model of Lu et al., since our methodology was conceptually very similar, we examined the patient cohorts and found that their cohort was more diverse regarding cancer type, while the dataset we used for this comparison contains only patients with metastatic GC [21]. This heterogeneity might account for the relative drop in predictive power. Examining this gene set, we have also noted that the shape of the averaged ROC curve is of particular interest (and is not wholly dissimilar from what the authors published). It suggests that above a certain regression value, the separation between responders and non-responders is quite clear, and only a relatively small group of the observations, those with the lowest regression value, are a hard-to-separate mixture of positives and negatives. If a variable identifying this mispredicted subset could be found, the prediction power on the remaining group would be particularly high, enabling the definition of a clinically relevant subset.

For this reason, we decided to further analyze the predictive power of Lu et al.’s onco-immunology gene set on the most clearly defined open-access ICI-treated GC cohort to date [7,21]. Using logistic regression, we defined the contribution of each gene to the model’s performance to predict ITR, then implemented the four most established clinical confounders from the article’s metadata: MSI, PD-L1 CPS, mutational load, and immune signature. Using the same logistic regression model penalizing these confounders in a second setting, we compared their predictive values with the genes’, arranged in a hierarchical order based on their coefficients. We revealed that the gene set published in the work of Lu et al. identified potential biomarkers that outperform traditional markers in multivariate analysis. M6PR has been associated with a potential anti-tumor role due to its ability to bind and degrade the mitogen insulin-like growth factor 2 (IGF2), but so far it was mentioned only in connection with breast cancer [40,41].

IDO1’s association with a response is a particularly controversial finding, due to the extensive literature on IDO1 for its immunosuppressive, pro-tumorigenic role in various solid tumors and GC [42,43,44,45]. In contrast, in fact, it was identified as part of an immunogenic signature during the transcriptomic characterization of the GC tumor microenvironment [46]. IDO1 inhibitors, like epacadostat, showed promise in early clinical trials by enhancing the effects of Programmed cell death protein 1 (PD-1) inhibitors. However, the phase III trial (ECHO-301/KN-252) combining epacadostat with pembrolizumab in metastatic melanoma did not meet its goals, leading to the termination of several phase III trials including one for endometrial cancer (NCT03310567) suggesting a much more complex biological background for the immune checkpoint, strengthening the argument behind the generalizable behavior of these biomarkers across multiple cancer types [47].

In contrast with IDO1, TNFSF9 was overwhelmingly reported as a potent T-cell activator with a positive effect on survival and immunotherapy outcomes [48,49,50,51], where epigenetic regulation plays a central role [52]. NRP1 has been described lately as a potential target for immune checkpoint inhibition (Chuckran et al., 2020), specifically due to its association with Treg (regulatory T) cells in the tumor microenvironment (Chuckran et al., 2021) [53,54]. VCAM1 and MAGEA3 have been already demonstrated to be prominent tumorigenic factors in GC, potentially responsible for impaired ITR by several studies [55,56,57,58].

In cancer biomarker research, overall survival (OS) stands as the definitive endpoint because it directly and unmistakably gauges the primary aim of cancer therapy: life extension. OS offers a straightforward evaluation of a treatment’s impact, unaffected by the biases or effects of subsequent treatments that can influence response rates or progression-free survival [59]. Thus, even in the absence of such data from large cohorts of IT-treated GC patients, we aimed to investigate survival in the case of the described individual biomarkers identified for GC ITR. The interactive web-based Kaplan–Meier plotter tool offers such an opportunity to use open-access datasets for GC patients who underwent surgical resection and/or chemotherapy [27]. We confirmed the prognostic role of the expression of M6PR and IDO1 in increasing, and for NRP1 and MAGEA3 in decreasing, OS on a large multi-cohort dataset of GC patients. However, it is important to note that neither response for OS nor OS for response can be unequivocally accepted as surrogate outcome measures due to potentially distinct underlying biological mechanisms.

The fact that our benchmark methods of using traditional clinical biomarkers and dimensionality reduction by PCA outperformed nearly all examined gene sets, except for that of Lu et al., advises caution when applying methodologies not primarily designed for the prediction of ITR for this purpose, even if the biological background suggests a connection between clinical benefit and the particular effect studied. Further research on larger datasets focused specifically on the prediction of responder status is necessary before real-world applications can be considered.

## 5. Conclusions

Machine learning methods hold the potential to improve ICI treatment outcome prediction. Using 5-fold cross-validation and LR, we verified the predictive power of applied ML algorithms on RNA signatures, on a well-defined ICI-treated GC dataset. In two cases our method has outperformed the published results. Interestingly, a study using a selected gene set (immuno-oncology panel) with AUC = 0.83 was the only one that outperformed TCB biomarkers such as MSI, PD-L1, or immune signature (AUC = 0.8) and the PCA (AUC =0.81) results. Based on our meta-analysis, we recommend further investigation and experimental validation in the case of M6PR, IDO1, NRP1, and MAGEA3 expression based on their strong predictive power in GC ITR. Well-designed studies with larger sample sizes and nonlinear ML models might help improve future biomarker studies.

## Figures and Tables

**Figure 1 ijms-26-05937-f001:**
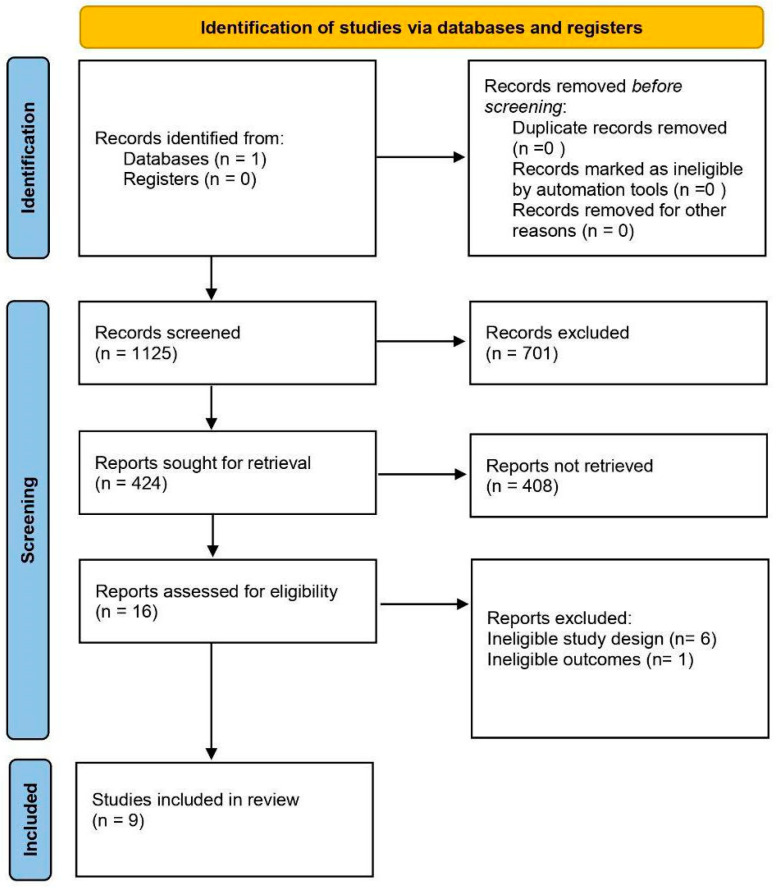
PRISMA flow chart of the search methodology for identifying relevant papers. A search of PubMed was conducted to identify studies using machine learning algorithms to predict immunotherapy efficacy in gastrointestinal cancers. Sixteen studies were assessed against eligibility criteria; six were excluded because of an ineligible study design and one because of an ineligible outcome.

**Figure 2 ijms-26-05937-f002:**
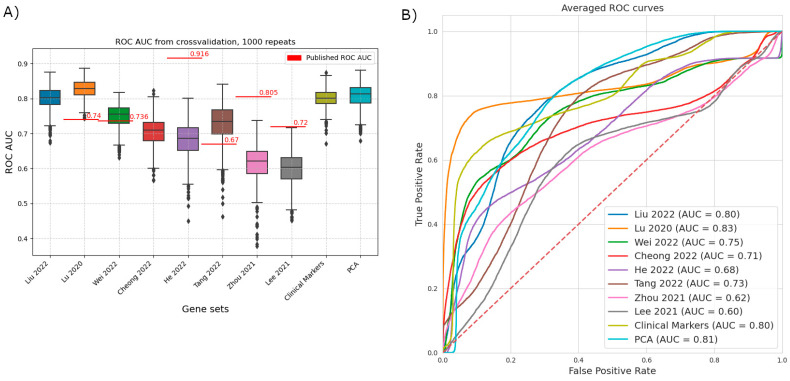
Comparative analysis of the predictive performance of previously published gene signatures using a uniform machine learning framework [1,7,13,22,23,24,25,26]. Data for the gene signatures highlighted by the different publications was extracted from the dataset published in Kim 2018, including the outcome variable response to immunotherapy [21]. Five-fold cross-validation with a stratified split was repeated 1000 times. The dataset is from Kim 2018, and the outcome is the response to immunotherapy [21]. (**A**) ROCAUC values of the individual repeats. Not all signatures retain high predictive power on the reference dataset. The center line and the sides of the box are the 25th, 50th, and 75th percentiles. Whiskers extend to the maximum and minimum values, excluding outliers (marked as circles, individually). Outliers are defined as values that are further from the sides of the box than 1.5× the interquartile range. Originally published AUC values are represented as red lines. Liu 2022 and Cheong 2022 did not publish ROC AUC values on a validation set [1,22]. (**B**) Averaged ROC curves, summarizing the results of all folds and repeats. TPR and FPR values are averaged for all values of the running threshold. Of particular note is the shape of the curve based on Lu 2020 (orange), where there is a threshold with relatively high TPR, but still low FPR [7].

**Figure 3 ijms-26-05937-f003:**
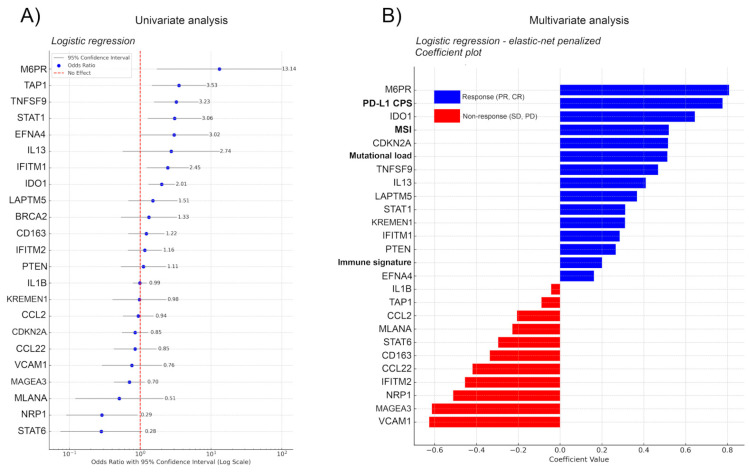
(**A**) Uni- and (**B**) multivariate regression analysis for genes in the immuno-oncology panel of Lu et al. [7]. Logistic regression with 5-fold cross-validation on the Kim et al. sequencing data analyzed the predictive power of individual genes in the immuno-oncology set of Lu et al. for immunotherapy response (ITR), focusing on Odds Ratio (OR) [7,21]. Coefficient values and classification accuracy are shown in Appendix A. Univariate analysis highlighted genes M6PR, TAP1, TNFSF9, STAT1, EFNA4, IL-13, IFITM1, and IDO1 as significant predictors of response (OR > 2) and STAT6, NRP1, MLANA, MAGEA3, and VCAM1 as predictors of non-response (OR < 0.8). Accuracy varied, with TNFSF9 and IDO1 reaching 93% and 86%, respectively, while EFNA4, IL-13, and M6PR exhibited lower accuracy (≤71%). Incorporating clinical covariates (MSI, Mutational load, PD-L1 CPS) through elastic-net-penalized logistic regression identified PD-L1 CPS and MSI as top predictors. M6PR, IDO1, CDKN2A, TNFSF9, and IL-13 emerged as significant genes affecting ITR, with M6PR surpassing established clinical markers. For non-response, VCAM1, MAGEA3, and NRP1 were strong predictors in the multivariate analysis. The model performed with an average AUC of 0.864, accuracy of 88.05%, and an average precision of 80.83%, indicating a relatively good predictive ability of the model, especially considering the accuracy and the balance between sensitivity and specificity as reflected by the AUC.

## Data Availability

All data used in our meta-analysis is open-access and available at url: https://doi.org/10.5281/zenodo.15206082.

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
