# Peer review of "Predicting Immunotherapy Efficacy with Machine Learning in Gastrointestinal Cancers: A Systematic Review and Meta-Analysis"

_ijms, 2025, doi:10.3390/ijms26135937_

Round 1

Reviewer 1 Report (New Reviewer)

Comments and Suggestions for Authors

Thank you for submitting your manuscript evaluating machine learning algorithms for predicting immune checkpoint inhibitor success in gastrointestinal cancers. I have reviewed your work with interest.

Your paper provides a valuable service to the field by systematically comparing various studies that employ machine learning approaches to predict immunotherapy response. The comprehensive analysis of nine eligible studies spanning pan-cancer, gastric cancer, and colorectal cancer contexts offers readers a useful overview of current research in this area.

While the manuscript has merit, I have two major concerns regarding your methodology:

  1. Feature Selection Methodology: Your use of elastic nets for feature selection is problematic. I strongly recommend reviewing the paper by Liu et al. (2023) published in Mathematics (https://www.mdpi.com/2227-7390/11/17/3738), which demonstrates significant limitations of this approach for high-dimensional biological data. The paper shows that elastic nets can fail to capture important features when dealing with complex genomic datasets. Superior feature selection methods are currently being developed (though not yet published) that could substantially improve your analysis. I encourage you to consider alternative approaches that better handle the complexity and high dimensionality of RNA expression data.
  2. Validation Using Logistic Regression: Your validation approach using logistic regression raises serious concerns about the robustness of your findings. As noted in your Figure 2 description, you compared gene sets "using a uniform methodology" with logistic regression. This approach fundamentally misses non-linear combinatorial effects that are likely critical in immune response prediction. Immune system interactions with cancer cells involve complex networks with numerous non-linear relationships that simple logistic regression cannot adequately capture. A more sophisticated validation approach incorporating non-linear models would provide a more accurate assessment of the predictive capabilities of these gene signatures.

Additional suggestions for improvement:

  1. The manuscript contains redundant sentences and grammatical issues that require attention, particularly in the descriptions of verification cohorts and TCGA datasets.
  2. Consider expanding on why certain ML algorithms performed better than others in your validation cohort.
  3. The relatively small sample size (n=45) for validation raises questions about statistical power and generalizability. A discussion of these limitations would strengthen your paper.
  4. Your paper would benefit from a more structured presentation of the verification cohorts, perhaps in a clear table summarizing all cohorts by cancer type, treatment received, and data type. Please...it was tough to follow. Find some way of presenting the information about the studies you reviewed and their scores. The tables are messy and difficult to read. Some other format...I do understand that this is challening but a small effort here could make a big difference. 
  5. A critical discussion on the comparative advantages of classification versus regression models in this specific context would add significant value.

Overall, while your manuscript makes a contribution to the field, addressing the major methodological concerns regarding feature selection and validation approaches would substantially strengthen its scientific rigor and impact. I recommend major revisions to address these issues before consideration for publication.

Some additional info for you: 

I'd recommend focusing on ensemble methods like Random Forest and Gradient Boosted Trees (like XGBoost, LightGBM, or CatBoost) combined with recursive feature selection approaches. Here's why these would be superior to logistic regression with elastic nets:

For feature selection:

  1. Recursive Feature Elimination (RFE) - This approach recursively removes features and builds a model on the remaining features, providing a ranking of features based on importance.
  2. Permutation importance - This method measures the decrease in model performance when a feature's values are randomly shuffled, identifying truly predictive features.
  3. SHAP (SHapley Additive exPlanations) values - These provide a unified measure of feature importance based on game theory principles.

For the modeling approach:

  1. Random Forest - Offers excellent performance for high-dimensional biological data as it:
    • Naturally handles non-linear relationships and feature interactions
    • Provides built-in feature importance measures
    • Is robust to overfitting when properly tuned
    • Maintains good performance even with correlated predictors
  2. Gradient Boosted Trees (XGBoost/LightGBM/CatBoost) - Often achieve state-of-the-art results because they:
    • Sequentially correct errors of previous trees
    • Handle complex non-linear patterns exceptionally well
    • Include regularization parameters to prevent overfitting
    • Offer flexibility with different loss functions

For validation, I'd suggest:

  • Nested cross-validation to avoid selection bias
  • Performance evaluation across multiple metrics (AUC, precision-recall curves, F1 score)
  • Calibration assessment of predicted probabilities

These approaches would better capture the complex, non-linear relationships in immune system interactions with cancer cells and provide more reliable feature selection than elastic nets, especially for the high-dimensional data typical in RNA expression studies.

Comments on the Quality of English Language

1) Please do not use the term "overperform". The term is outperform or improve.

The English is not terrible but it could use a read over from a native speaker to fix issues and better structure the language. It is not terrible at all however. 

You reference CGNN without defining it in a table. Please specify that you meant convolutional graph neural network. 

Author Response

  1. Summary

We thank Reviewer 1 for the thoughtful and constructive feedback on our manuscript. We appreciate the opportunity to revise our work and address the raised concerns. Below, we provide point-by-point responses to the reviewer’s comments. All changes made in the manuscript have been clearly marked.

  1. Point-by-point response to Comments
    Comment 1: Feature Selection Methodology: Your use of elastic nets for feature selection is problematic. I strongly recommend reviewing the paper by Liu et al. (2023) published in Mathematics (https://www.mdpi.com/2227-7390/11/17/3738), which demonstrates significant limitations of this approach for high-dimensional biological data. The paper shows that elastic nets can fail to capture important features when dealing with complex genomic datasets. Superior feature selection methods are currently being developed (though not yet published) that could substantially improve your analysis. I encourage you to consider alternative approaches that better handle the complexity and high dimensionality of RNA expression data.”

Response1:
We thank the reviewer for this important remark and the hint that superior methods are ahead of publication. A comparison of different variable selection methods would go beyond the scope of our meta-analysis since most included studies utilized only standard libraries from R (see Table 2), therefore we decided to remain in this context. However for raising awareness we included a reference on the topic on over-selecting features of the elastic net approach mentioned in the comment into the revised manuscript. A similar result on variable selection to Liu et al. was obtained by us using a greedy variable selection algorithm. The leave-one-out cross-validation governed variable selection was more characteristic compared to the leave-many-out one which seemed to select too many different features [1]

Despite our data was not high-dimensional we counterchecked our elastic net analysis with a 1000 times repeated stratified 5-fold cross validated logistic regression modelling and recursive feature elimination (rfe). For each training segment the rfe selected 10 features based on the assigned weights. TFNSF9, IDO1, M6PR were in the intersection of genes selected by the elastic net approach and rfe.

[1] Király P, Tóth G. Being Aware of Data Leakage and Cross-Validation Scaling in Chemometric Model Validation. J Chemom 2025;39:e70026. https://doi.org/10.1002/cem.70026.

The new part (Page 13, lines 43-48): 

“First we fitted an elastic-net penalized logistic regression model, with a custom penalty mix, penalizing only the clinical covariates. As this data is not high-dimensional, we chose this standard method for simplicity. We are aware that for full transcriptomic datasets, the effectiveness of LASSO and elastic net variable selection is being questioned [Liu 2023].”

Comment 2: “Validation Using Logistic Regression: Your validation approach using logistic regression raises serious concerns about the robustness of your findings. As noted in your Figure 2 description, you compared gene sets "using a uniform methodology" with logistic regression. This approach fundamentally misses non-linear combinatorial effects that are likely critical in immune response prediction. Immune system interactions with cancer cells involve complex networks with numerous non-linear relationships that simple logistic regression cannot adequately capture. A more sophisticated validation approach incorporating non-linear models would provide a more accurate assessment of the predictive capabilities of these gene signatures.”

Response 2:

The reviewer refers to our comparison of gene sets as “validation”. While it is true that on the surface it looks similar, we did not call it as such on purpose.

Throughout the manuscript we refer as validation only to statistical model validation. The models themselves were assessed numerically on robustness by cross-validation. Since a model built on n=45 objects is easily overfitted [1], we decided not to use train/test splitting for external validation rather to use all objects and validate the models only by cross-validation which in such cases can be thought of a “suboptimal simulation of test set validation” [2]. Also an external validation without modeling would have been only possible if all models corresponding to the gene signatures would have been published.

Our aim was not to validate the findings in the other articles, but to provide the best available comparison between them and the well established clinical markers. Thus, while the reviewer’s comments would be pertinent and quite important in an actual validation study, their applicability to our work is limited.

We agree with Reviewer 1 that logistic regression as a linear method has the disadvantage not to catch nonlinearities of high dimensional transcriptomic data which are rather approximated by nonlinear methods like neural networks and ensemble methods. We also note that signatures are only a subset of the original transcriptomic data therefore high dimensionality (p>n) is less of relevance, the largest subset has dimensions of n=45 and p=33. However all included studies but one used linear methods for modelling therefore we decided to compare the signatures by a linear method, for simplicity logistic regression.

For true high dimensional data we recently demonstrated the use of ensemble methods [3,4].

[1] Király P, Kiss R, Kovács D, Ballaj A, Tóth G. The Relevance of Goodness-of-fit, Robustness and Prediction Validation Categories of OECD-QSAR Principles with Respect to Sample Size and Model Type. Mol Inform 2022;41:e2200072. https://doi.org/10.1002/minf.202200072.

[2] Esbensen KH, Geladi P. Principles of Proper Validation: use and abuse of re-sampling for validation. J Chemom 2010;24:168–87. https://doi.org/10.1002/cem.1310.

[3] Paál Á, Dora D, Takács Á, Rivard C, Pickard SL, Hirsch FR, et al. Roles of Annexin A1 Expression in Small Cell Lung Cancer. Cancers 2025;17:1407. https://doi.org/10.3390/cancers17091407.

[4] Dora D, Kiraly P, Somodi C, Ligeti B, Dulka E, Galffy G, et al. Gut metatranscriptomics based de novo assembly reveals microbial signatures predicting immunotherapy outcomes in non-small cell lung cancer. J Transl Med 2024;22:1044. https://doi.org/10.1186/s12967-024-05835-y.

Comment 3: “The manuscript contains redundant sentences and grammatical issues that require attention, particularly in the descriptions of verification cohorts and TCGA datasets.”

Response 3:
We thank the reviewer for this detailed observation. We acknowledge that the presentation of the verification cohort and TCGA-related information was dense and may have challenged readability.

The section in question (Results, paragraph 2) aimed to present a high volume of cohort-level details in a compact narrative format. Each sentence provided discrete and non-overlapping information: tumor types across training and validation cohorts, treatment, and cohort sources. Thus, while the text may have appeared overwhelming, it was not redundant in content.

Nonetheless, we recognize that the long paragraph structure made it harder to follow. In response to the reviewer’s helpful suggestion:

  1. We edited and clarified the original text to improve sentence structure and eliminate stylistic inconsistencies. (Page 5., lines  4-27, Study characteristics section.)

New part: “Regarding verification cohorts, most studies included GC (n=8), melanoma (n=6), and urothelial cancers (n=6), with additional cohorts covering CRC (n=4), esophageal (n=3), and lung cancers (n=3). Other cancers were also represented in smaller numbers (n=2) (as detailed in Table 1).One study also includes data from normal tissue in the verification cohort (He et al., 2022)

Immune checkpoint inhibitor (ICI)-treated cohorts primarily involved gastric cancer (n=6), melanoma (n=5), and urothelial cancer (n=5), with additional inclusion of colorectal (n=1), esophageal (n=1), lung (n=2), and kidney (n=1) cancers.

Two studies (Lu et al., 2020; Zhao et al., 2023) included immunotherapy-treated patients directly in their training cohorts, specifically with anti-PD-1, anti-PD-L1, and anti-CTLA-4 therapies.

In the verification cohorts, anti-PD-1 or PD-L1 inhibitors were used in five studies each, and anti-CTLA-4 in five as well. While most TCGA datasets were from non-immunotherapy-treated cohorts, two datasets contain immunotherapy-exposed patients: TCGA-SKCM (melanoma; 24 patients) and TCGA-BLCA (urothelial cancer; 2 patients), both treated with anti-PD-1/CTLA-4.

All studies analyzed RNA-seq data from human tumor tissues, with He et al. (2022) also incorporating mouse colon cancer data. Survival data were available in all studies: overall survival (OS, n=8), progression-free survival (PFS, n=4), and recurrence-free survival (RFS, n=2). Response outcomes included response rate (RR, n=6) and durable clinical benefit/no durable benefit (DCB/NDB, n=1). Gene signatures were initially tested in either chemotherapy (n=7) or immunotherapy (n=2) training cohorts. Classification models were used in five studies, and regression models in four. All models were evaluated on immunotherapy-treated cohorts.”

  1. We made several formatting adjustments to improve readability and visual structure. Specifically, we removed one column from Table 1 and instead incorporated that information into the Study characteristics section of the manuscript. (Page 4, Line 27. New Table 1: page 6)

New part: “All models were evaluated on immunotherapy-treated cohorts.”

Potentially redundant or overlapping content was consolidated in the explanatory caption of Table 1 to enhance clarity and facilitate interpretation.

New part:

 “*IT in melanoma anti-PD1/anti-CTLA4 26/442, IT in urothelial cancer anti-PD1/anti-CTLA4 2/411.

1Tumour types were not further specified in the study.

2 Gastric, colorectal, esophageal, head and neck, bladder, breast, cervical, bile duct, lymphoma,  kidney, liver, lung, ovarian, pancreatic, prostate, melanoma, thyroid, uterine cancers, neuroendocrine tumors, glioma, glioblastoma, lymphoma, mesothelioma, sarcoma, thymoma.

3 Gastric, colorectal, esophageal, bile duct, pancreatic, liver, lung, nasopharyngeal, thyroid, bladder, kidney, prostate, uterine, ovarian, cervical, melanoma, skin, brain, head and neck, oral, breast, bone cancers, lymphoma, chronic lymphocytic leukemia, chronic myeloid disorders, lymphoproliferative syndrome, rare pancreatic tumors, glioma, glioblastoma, pediatric brain tumors, neuroendocrine tumors, thymoma, sarcoma, mesothelioma.

4 Ovarian, breast, leukemia, multiple myeloma, liver, kidney, bladder cancers

IT: Immunotherapy, ML: Machine learning, CRC: Colorectal, GI: Gastrointestinal, OS: overall survival, PFS: progression-free survival, RFS: recurrence-free survival, DCB/NDB: durable clinical benefit/no durable benefit, RF: random forest, XGboost: extreme gradient boosting, SVM: support vector machine, GCNN: convolutional graph neural network”

  1. Table 2 was transposed to allow for better horizontal readability, and we also reduced the font size within the tables to enhance overall clarity. No content-related changes were made to the data presented. These adjustments aim to improve the accessibility of the tables without altering their informational value.  (Pages 7-8.)

We believe these changes improve the manuscript’s clarity and accessibility.

Comment 4:Consider expanding on why certain ML algorithms performed better than others in your validation cohort.”

Response 4:
If we understand correctly, the question refers to comparing our own results to each other in the comparison test using the Kim dataset. As the difference between these is in the underlying gene set used (the methodology is otherwise identical), to answer this question, we would have to investigate the gene sets from a genetic and molecular biological point of view, which would be the topic of another entire review. If we misunderstood the question, we kindly ask the Reviewer to further specify their question.

Comment 5:The relatively small sample size (n=45) for validation raises questions about statistical power and generalizability.”

Response 5:

Yes, we agree with the reviewer, this concern is something we also emphasize in the conclusion, and one of the reasons we aimed to do a comparison of the highlighted gene sets and not a validation of the published models. Scarcity of immunotherapy cohort data seems to lead the field to indirect modelling approaches, trading the problem of small sample size for lower relevance of data (e.g. using TCGA without a clear ITR outcome variable). Please also note the comment and reference on overfitting at low sample sizes in the answer to the major concern #2. For models at low sample sizes it is essential to rigorously validate at all three levels, internally for goodness of fit, robustness and externally for predictivity to avoid overfitted models.

Comment 6: “Your paper would benefit from a more structured presentation of the verification cohorts, perhaps in a clear table summarizing all cohorts by cancer type, treatment received, and data type. Please...it was tough to follow. Find some way of presenting the information about the studies you reviewed and their scores. The tables are messy and difficult to read. Some other format...I do understand that this is challening but a small effort here could make a big difference. “

Response 6:
We thank the reviewer for underscoring the importance of a clear presentation of verification cohorts. After revising our tables as described in response 3, we believe that all cohort details are now presented in a highly accessible format. In particular:

  • Table 1 has been streamlined by removing the column “Model evaluated on IT Cohort,” which for all nine studies contained the same entry (“1") and therefore did not offer any comparative insight. That detail has been incorporated into the main text. (Page 5)
  • Table 2 has been transposed and reformatted, and its font size reduced to allow for comfortable reading of cohort characteristics and performance metrics. (Page 7-8)
  • Table 3 has been edited to the same format (Page 9)

These targeted formatting adjustments ensure that readers can readily extract the requested information without introducing a redundant summary table. We therefore respectfully submit that the revised Tables 1 and 2 now fully satisfy the reviewer’s request for structure and readability, while avoiding unnecessary duplication of content.

Comment 7:A critical discussion on the comparative advantages of classification versus regression models in this specific context would add significant value.”

Response 7:
We appreciate the reviewer’s suggestion and agree that exploring the comparative advantages of classification versus regression models could provide valuable insights. While certainly valuable, diving into a detailed, general comparison of two major types of prediction methodologies would be out of scope for this article.

Comment 8: “Please do not use the term "overperform". The term is outperform or improve. 

The English is not terrible but it could use a read over from a native speaker to fix issues and better structure the language. It is not terrible at all however.  

You reference CGNN without defining it in a table. Please specify that you meant convolutional graph neural network”

Response 8:
Thank you for your comment. We have carefully revised the manuscript and corrected all instances where the term "overperform" was used. It has now been replaced. We have also fixed all typos found.
We now specify “Convolutional Graph Neural Network” (CGNN) at its first mention and include it in Table 1 (Page 5). 

  1. Response to Comments on the Quality of English Language

Point 1: “The English is not terrible but it could use a read over from a native speaker to fix issues and better structure the language. It is not terrible at all however. 

Response 1: We thank the reviewer for the suggestion regarding English language clarity. After carefully reviewing the manuscript,  we made targeted adjustments to improve textual coherence and flow, which we believe enhance overall readability. 

Reviewer 2 Report (New Reviewer)

Comments and Suggestions for Authors

This paper offers a well-realized systematic review and meta-analysis assessing how machine learning methods can improve prediction of immunotherapy outcomes in gastrointestinal cancers. The available literature has been thoroughly researched, withclearly defined inclusion criteria and a special focusin the analysis on specific RNA expression signatures. One of the most positive aspects of this study is in its methodology; a unified logistic regression framework on a standardized gastric cancer cohort has been used for analytic purposes, adding a supplemental layer of validation and, therefore, reinforcing the validity of the results.

Their results indicate that certain expressionsignatures generated by machine learning can show better predictive performance compared to traditional biomarkers like PD-L1 and MSI. The authors also highlight individual genes such as IDO1 and M6PR as potential biomarkers, based on both predictive models and prognostic analysis. These shows as particularly relevant findings, considering howeligibility criteria and their optimal definition for immunotherapy remains a clinical challenge. Though promising, it’s worth mentioning that the study indeed presents some limitations: databases analyzed are either small or heterogeneous. This somewhat limits the potential for wide general applications of the results and should lower expectations for a hurried, immediate clinical translation.

Overall, the review is rigorously designed and gives a potentially valuable contribution into an interesting,almost futuristic topic such as the interplay betweenmachine learning models and oncology. Given the strengths presented above, I recommend the manuscript for publication after minor revisions.

Author Response

  1. Summary
    Thank you very much for taking the time to review this manuscript. Please find the detailed responses below and the corresponding revisions in track changes in the re-submitted files.
  2. Point-by-point response to Comments
    Comment 1: “Though promising, it’s worth mentioning that the study indeed presents some limitations: databases analyzed are either small or heterogeneous. This somewhat limits the potential for wide general applications of the results and should lower expectations for a hurried, immediate clinical translation.“

Response 1. Yes, we agree with the reviewer, this concern was emphasized in the “Discussion”, paragraph 2 and in the closing remark of the “Conclusion”, which points to anticipation of more gastrointestinal immunotherapy datasets in future. Scarcity of immunotherapy cohort data seems to lead the field to indirect modelling approaches, trading the problem of small sample size for lower relevance of data (e.g. using TCGA without a clear ITR outcome variable). For models at low sample sizes it is essential to rigorously validate at all three levels, internally for goodness of fit, robustness and externally for predictivity to avoid overfitted models. Furthermore, there were only two studies where the response to immunotherapy was directly modeled (the modeling approaches are described in detail in the Modeling section). Other models focus on a predefined biological context, and therefore should be interpreted within that specific context, limiting generalization. This limitation is addressed in the final paragraph of the Discussion section. 

  1. Additional clarifications:
    We revised the manuscript text and reformatted Tables 1–3 to improve clarity, consistency, and readability. We believe these changes improve the manuscript’s clarity and accessibility.

This manuscript is a resubmission of an earlier submission. The following is a list of the peer review reports and author responses from that submission.

Round 1

Reviewer 1 Report

Comments and Suggestions for Authors

This systematic review and meta-analysis evaluate the predictive performance of ML models by analyzing nine studies that applied ML techniques to tumor RNA expression profiles of ICI-treated GI cancer patients. The findings highlight the variability in predictive power across different ML approaches, with one immuno-oncology gene panel achieving the highest predictive accuracy, outperforming conventional clinical biomarkers. These results underscore the need for further validation and larger, well-designed studies to optimize ML-driven biomarker selection in immunotherapy. The manuscript is extremely well written and detailed. I only have a few minor comments:

1. Authors excluded and removed a lot of reports - can you clarify why these many records were included in initial screening and then rejected from the study? In other words, what were the conditions you used to eliminate these?

2. Data availability statement - if I understood correctly, Authors have also used publicly available data for some analysis? If yes, please include the links to it here.

3. For figure 3, could the figure itself or the fonts within it be slightly enlarged, for easier readability.

4. Please add grids to Figure 2.

Author Response

„This systematic review and meta-analysis evaluate the predictive performance of ML models by analyzing nine studies that applied ML techniques to tumor RNA expression profiles of ICI-treated GI cancer patients. The findings highlight the variability in predictive power across different ML approaches, with one immuno-oncology gene panel achieving the highest predictive accuracy, outperforming conventional clinical biomarkers. These results underscore the need for further validation and larger, well-designed studies to optimize ML-driven biomarker selection in immunotherapy. The manuscript is extremely well written and detailed. I only have a few minor comments:

„1. Authors excluded and removed a lot of reports - can you clarify why these many records were included in initial screening and then rejected from the study? In other words, what were the conditions you used to eliminate these?”

Thank you for pointing this out. We have employed an expansive approach to the initial keyword search not to miss any relevant publications but also necessitating the removal of 701 matches that did not fit our inclusion criteria based on their abstracts. The full definition of search keywords can be found in Supplementary File F1. The inclusion/exclusion criteria are detailed in the Materials and Methods section under Study Selection.

We supplemented the relevant section to ensure clarity:

“After screening the abstracts based on the inclusion/exclusion criteria, 701 were excluded due to irrelevance.”

„2. Data availability statement - if I understood correctly, Authors have also used publicly available data for some analysis? If yes, please include the links to it here.”

On the comment: We used the open access, processed data from Kim et al. [1] for our calculations, which is available from the Tumor Immune Dysfunction and Exclusion (TIDE) framework [2]. The raw data can be found in the European Nucleotide Archive where it was registered by Kim et al. under accession PRJEB25780.

[1] Kim et al., Nat Medicine 2018, https://doi.org/10.1038/s41591-018-0101-z

[2] Jiang et al., Nat Medicine 2018, https://doi.org/10.1038/s41591-018-0136-1

The corresponding data availability statement was corrected:

“Data availability statement: All data used in our meta-analysis is open-access and available at url: http://tide.dfci.harvard.edu/.”

„3. For figure 3, could the figure itself or the fonts within it be slightly enlarged, for easier readability.

As requested, we increased font size for gene names in Figure 3.

„4. Please add grids to Figure 2.”

Thank you for the comment, grids were added to Figure 2 for better readability.

Reviewer 2 Report

Comments and Suggestions for Authors

Szincsaket al, reviewed some of the papers reported immunotherapy efficacy in Gastrointestinal cancers by means of machine learning. Although the topic is interesting, the authors very confusingly present their data and conclusions.

  • The text is, in general, cryptic and occasionally repetition of the table in most places which makes it difficult to find a flow in the manuscript.
  • Sometimes you can find abbreviation that have not been properly shown (e.g. can reliably predict response to ICI (ITR)).
  • Authors do not discuss or even include potential technical differences among the selected studies which can impact the performance of the models used in this study.
  • Also, the authors do not acknowledge the differences of the sample size and their impact on the outcome of the used statistical framework.
  • What do you mean by this: “Two authors independently extracted data and resolved conflicts by active discussion” ?
  • The methodology is very poorly describing what has been done and current explanation make it impossible to first understand the exact statistical framework and second to be replicated by others. No statistical model no repository shared to see how the analyses were done. By the way in which program did the authors perform the analysis? R? Python? SAS? Matlab???
  • There are a lot of occasions which lead into confusions in the presentation of the data:
    • Figure2: “The data is from (kim et al., 2018)….” Panel A shows ROC-AUC in different studies which have used different methodology and they have their own dataset. I cannot make any conclusion from this representation?!
    • Authors are also explaining the boxplot, outliers, ….. Is it necessary to give such explanation in a scientific manuscript?
Comments on the Quality of English Language

The text requires major revision for English and how the data and results are represented.

Author Response

„Szincsak et al, reviewed some of the papers reported immunotherapy efficacy in Gastrointestinal cancers by means of machine learning. Although the topic is interesting, the authors very confusingly present their data and conclusions.”

„The text is, in general, cryptic and occasionally repetition of the table in most places which makes it difficult to find a flow in the manuscript.”

The Reviewer’s comment is well-taken. Since one of the results of the systematic review is the comparison of study characteristics, this is not redundant information and is also reiterated in the text. We have revised the first section to be less detailed while better summarizing the findings.

„Sometimes you can find abbreviation that have not been properly shown (e.g. can reliably predict response to ICI (ITR)).”

On the comment: We explicitly stated ITR in the abstract. “The outcomes included were response to immunotherapy (ITR)”.

The sentence was corrected for better clarity:

“One of the significant issues for Immune checkpoint inhibitor (ICI) - efficacy is the identification of markers across cancers that can reliably predict immunotherapy response (ITR)”

„Authors do not discuss or even include potential technical differences among the selected studies which can impact the performance of the models used in this study.”

 On the comment: Technical differences of the modelling steps were extensively discussed in sections Source of transcriptomic data, Variable selection, Modeling and Validation. Biological differences of the training cohorts are displayed in Table 1. and discussed briefly in section Study characteristics. All of these factors have an effect on model performance. It is well known that hyperparameter optimization also significantly influences model performance, unfortunately the documentation of it was not accessible in most of the selected studies therefore we discarded its discussion.

„Also, the authors do not acknowledge the differences of the sample size and their impact on the outcome of the used statistical framework.”

On the comment: Within the statistical framework a comparison of model performance is not possible due to heterogeneity of used modelling methods and datasets. Although figures of merit are presented in Table 2. we can not conclude whether one model is superior to other ones in predicting efficacy of immunotherapy. Generally in chapter "Source of transcriptomic data" of our paper we draw a conclusion regarding possible overfitting in terms of relative degrees of freedom of the model based on the comparison of the sample sizes in Table 2. In chapter “Validation” we emphasize that robustness was not assessed by some authors although this could indicate overfitting when the model was trained at small sample size. In the discussion we point out the scarcity of immunotherapy cohorts, therefore these small sample size datasets should also be used for modelling since efficacy can be predicted directly. In Table 2. the sample size dependence study of Zhao et al. for validation purposes was emphasized.

„What do you mean by this: “Two authors independently extracted data and resolved conflicts by active discussion” ?”

On the comment: Based on the “Cochrane Handbook for Systematic Reviews of Interventions” in section 5.5.5: “When more than one author extracts data from the same reports, there is potential for disagreement. After data have been extracted independently by two or more extractors, responses must be compared to assure agreement or to identify discrepancies. An explicit procedure or decision rule should be specified in the protocol for identifying and resolving disagreements. Most often, the source of the disagreement is an error by one of the extractors and is easily resolved.”

Thus, discussion among the authors is a sensible first step.

We have refined the description accordingly:

“Two authors independently extracted data, and any disagreements or discrepancies were discussed and resolved through the assessment of study eligibility.”

„The methodology is very poorly describing what has been done and current explanation make it impossible to first understand the exact statistical framework and second to be replicated by others. No statistical model no repository shared to see how the analyses were done. By the way in which program did the authors perform the analysis? R? Python? SAS? Matlab???”

A more detailed description of the statistical analysis is provided now at the end of the Statistical Analysis section in Materials and Methods:

“The scikit-learn library (Python) was employed to implement logistic regression models. The results of the logistic regression, PCA and ROC analyses, along with the calculated metrics, were compiled in a jupyter notebook which was placed at a Mendeley data repository to demonstrate reproducibility. Due to the small sample size, a 1000-fold stratified 5-fold split was used. To reduce the error due to chance, all models reported in the article were evaluated on identical splits which were created manually from random permutations of the samples. For full details see the attached jupyter notebook. The Kaplan-Meier (KM) Plotter online framework created by GyÅ‘rffy(GyÅ‘rffy, 2024; Nagy et al., 2021) was used to perform KM-analysis in assessing the prognostic role of key genes identified in the IT-treated GC dataset of Kim et al. The KM plotter is freely accessible at https://kmplot.com/analysis/.”

 „There are a lot of occasions which lead into confusions in the presentation of the data:

Figure2: “The data is from (kim et al., 2018)….” Panel A shows ROC-AUC in different studies which have used different methodology and they have their own dataset. I cannot make any conclusion from this representation?!

We revised the figure titled: “Comparative analysis of the predictive performance of previously published gene signatures using a uniform machine learning framework.”

And we added additional information to improve clarity:

“Data for the gene signatures highlighted by the different publications was extracted from the dataset published in (Kim et al., 2018), including the outcome variable response to immunotherapy. 5-fold cross-validation with a stratified split was repeated 1000 times.”

Authors are also explaining the boxplot, outliers, ….. Is it necessary to give such explanation in a scientific manuscript?”

On the comment: We fully agree with the reviewer that for trained statisticians this seems to be obsolete information while other readers are reminded of the box plot definition. Based on the authors’ experience it is necessary to define box and whiskers to prevent misunderstandings of the data representation. See e.g. the excellent wikipedia record on box plots (https://en.wikipedia.org/wiki/Box_plot), in section “Whiskers” different definitions for whiskers are listed.